# Acquisition and Retention Effects of Fundamental Movement Skills on Physical Activity and Health-Related Fitness of Children with Autism Spectrum Disorder

**DOI:** 10.3390/healthcare12131304

**Published:** 2024-06-29

**Authors:** Yu Xing, Haoyan Liu, Xueping Wu

**Affiliations:** 1School of Physical Education, Hainan University, Haikou 570228, China; liuhaoyan@hainanu.edu.cn; 2Hainan Provincial Key Laboratory of Sports and Health Promotion, Hainan Medical University, Haikou 571199, China; 3School of Physical Education and Training, Shanghai University of Sport, Shanghai 200438, China

**Keywords:** autism spectrum disorder, children, fundamental movement skills, physical activity, health-related fitness

## Abstract

This study adopted a quasi-experimental design to explore the effects of fundamental movement skill intervention on the acquisition and retention of physical activity levels and health-related fitness in children with autism spectrum disorder (ASD). In the experiments, 11 children received fundamental motor skill training (12 weeks, 60 min/session, 4 times/week), and 10 children maintained traditional physical activity. Assessments were performed using an ActiGraph GT3X+ accelerometer, health-related fitness pre–post intervention, and 1-month follow-up tests. The sedentary time during physical activity was significantly decreased (*p* = 0.01), and there were large changes in health-related physical fitness indicators, including significantly improved body composition (body mass index, F(1,19) = 8.631, *p* = 0.03, partial η^2^ = 0.312), muscle strength and endurance (sit-ups, F(1,19) = 3.376, *p* = 0.02, partial η^2^ = 0.151 and vertical jumps, F(1,19) = 5.309, *p* = 0.04, partial η^2^ = 0.218), and flexibility (sit and reach, F(1,19) = 36.228, *p* = 0.02, partial η^2^ = 0.656). Moreover, the follow-up tests showed that the children’s sedentary time continued to reduce, and the muscle strength and endurance (sit-ups, F(1,19) = 4.215, *p* = 0.01, partial η^2^ = 0.426) improved continuously after the intervention. Based on this study, actionable and regular fundamental movement skill programs can be provided in the future as an effective way to achieve the healthy development goals of children with ASD.

## 1. Introduction 

In the US, a report released by the Center for Disease Control and Prevention (CDC) in March 2023 showed that about 1 in 36 children has been identified as having ASD [1,2]. In China, according to a paper published in the core journal of the *Chinese Journal of Reproductive Health*, the prevalence of ASD, when screened in children aged 0 to 6 years in 2022, was 1.8% [3,4,5]. Referring to the guidelines of physical activity for children and adolescents, the physical activity compliance rate for children and adolescents with ASD in China is 26.79% [3]. Studies have shown that children with ASD face more health problems (e.g., gastrointestinal issues, sleep disturbances, sensory sensitivities, mental health conditions, and epilepsy) and challenges in communication, social interaction, behavior, motor skills, and daily activities compared to their neurotypical peers [5,6,7,8,9,10]. Although there are no significant differences in height, weight, BMI, and flexibility between ASD children and neurotypical children of the same age, significant differences exist in cardiorespiratory function and muscle strength [11,12]. Taking the VO_2max_ (maximum oxygen uptake) as an example, neurotypical children typically have VO_2max_ values ranging from 40 to 50 mL/kg/min, while children with ASD may have values closer to 30 to 35 mL/kg/min. This lower aerobic capacity indicates reduced endurance and greater difficulty sustaining physical activity [12,13,14,15,16]. These differences may reduce their willingness to participate in physical activities, impacting their physical health [13,14,15,16]. Therefore, exploring the relationship between physical activity, health-related fitness, and the mental and physical health status of children with ASD, as well as unveiling the impact mechanisms of different intervention strategies on these factors, holds significant practical importance for mitigating disease progression in children with ASD. It could help provide more effective support and intervention measures for children with ASD, thereby improving their quality of life and health status. 

Physical activity and health-related fitness can comprehensively reflect the condition and progression of children with ASD, offering a more holistic understanding of their health status and disease development level [17,18,19,20]. Physical activity is defined as any physical movement that results in energy expenditure caused by skeletal muscle contraction according to the American College of Sports Medicine [15]. Studies indicate that children with ASD engage in significantly less physical activity compared to their typically developing peers, a trend that is increasingly common worldwide. For example, typically developing children may average around 10,000 steps per day, while children with ASD often average between 5000 to 7000 steps per day. Additionally, the odds ratio for participation in organized sports among children with ASD versus typically developing children is approximately 0.5 to 0.7. These phenomena indicate the challenges that children with ASD face in achieving recommended levels of activity [17,18,19,20,21]. This disparity underscores the importance of physical activity in assessing the health and disease progression of children with ASD. Therefore, focusing on and intervening in the physical activities of children with ASD is crucial not only for their holistic health but also as a vital marker for monitoring and evaluating the progression of their condition. Furthermore, health-related fitness refers to the ability of the cardiovascular system, lungs, and muscles to perform at optimal efficiency. It is considered essential in the evaluation of the functional status of the human body and the risk control of developing chronic diseases [21,22,23,24,25], especially for individuals with ASD. For instance, research indicates that approximately 30% to 40% of children with ASD are overweight or obese, whereas the prevalence among the general pediatric population is around 20%. This higher prevalence of obesity in those with ASD not only affects physical health but also contributes to the overall challenges in managing health outcomes in this population [22,23,24]. Therefore, it is believed that a certain level of physical activity and fitness has a positive effect on controlling obesity and improving cardiopulmonary endurance. It can not only promote the suppression of the effects of ASD but also reduce the risk of multiple chronic non-communicable diseases (e.g., hypertension and diabetes) in adulthood [16,18]. 

Fundamental movement skills are recognized as important factors that significantly influence the physical activity and health-related fitness of children with ASD. These skills, which mainly include locomotor skills, objective control skills, and balance skills, form the basis for coordinating and utilizing the fundamental movements of the human body [14,26,27]. They are considered the cornerstone of an individual’s successful participation in organized or unorganized games, sports, and recreational activities, and are also regarded as a necessary prerequisite for future participation in more specialized or complex skills [19,20]. Fundamental movement skill interventions can be adapted to the motor development of children with ASD and further enhance the physical activity and health-related fitness levels in this population. Research shows that for children or adolescents with ASD, fundamental movement skill intervention effectively reduced the sedentary time and improved the low-intensity physical activity and isometric push up [15,27]. Two meta-analyses in these studies [13,20,21,22,23,24] explored the effect of different forms of physical activity interventions on children with ASD [13,20], including fundamental movement skills related to body composition (e.g., body mass index [21,22]), cardiovascular fitness [23], muscular strength/endurance, and flexibility [24]. The majority of studies found that fundamental movement skills could be considered fun by children and enhance the competitivity and cooperativity of motor activity, which promoted the participation of children with ASD, as well as improving their physical activity and health fitness in a more comprehensive way [13,20]. Furthermore, studies also show that children with ASD show different preferences for fundamental movement skills-based interventions. They usually like to dance, walk, and run to music, and some of them with strong athletic ability are also willing to practice ball games (e.g., table tennis) when it comes to using equipment [28]. Compared with a neurotypical population, children with ASD are often a disadvantaged and neglected group, with higher rates of morbidity, physical disability, and health problems [15]. To date, there are few studies that have established a correlation between various fundamental motor skill interventions and physical activity and the health-related fitness of children with ASD.

Furthermore, data sources and processing methods, as well as being precise, are critical when discussing the benefits of basic exercise interventions on physical activity and health-related fitness in children with ASD [21]. Ketcheson introduced Classroom Pivotal Response Teaching (CPRT) into motor skill intervention for the first time and found that the motor skills of children with autism were significantly improved. Notably, CPRT is defined as a teaching method that helps children with autism acquire core skills through the natural environment and the improvement of motivation levels, thus leading to a derivative impact on other behaviors [29,30]. The 4-week follow-up tests showed that the intervention effect was well maintained, which suggested the importance of fundamental movement intervention for the physical health of children with autism. Bo and Dong verified the effectiveness of a 2- and 9-week motor skill intervention on motor ability and the core symptoms of the intervention. They found that the overall performance of children with autism improved after the 2-month follow-up intervention [22,23]. To date, current studies lack follow-up testing of physical activity and health-related physical fitness after an intervention related to different fundamental motor skills, which causes difficulties in comprehensively and accurately evaluating the impact of the proposed intervention strategy on the condition of children with ASD. In this context, this study selected fundamental movement skills that meet their health needs as an intervention to investigate the acquisition and retention effects of this fundamental movement skills on physical activity and health fitness in children with autism through pre/post-evaluations and long-term follow-up tests. Notably, the retention effects refer to the sustained impact of the intervention observed at follow-up assessments conducted after the completion of the intervention period. Specifically, retention effects are measured by comparing the outcomes at the end of the intervention with those at subsequent follow-up points (e.g., 3 months and 6 months post-intervention) to evaluate the longevity and persistence of the intervention’s benefits. Furthermore, the hypothesis of this study is that the fundamental movement skill intervention has a positive impact on physical activity and health-related fitness in children with ASD. 

## 2. Materials and Methods

### 2.1. Participants

This study adopted a quasi-experimental design, was conducted in accordance with the guidelines of the Declaration of Helsinki, and was approved by the Ethics Committee of Shanghai University of Sport.

Schools for children with intellectual disabilities in Shanghai and Wuhu City, Anhui Province, were selected as recruitment sites. The following criteria were used for recruitment and inclusion or exclusion in this study: (1) children aged 7–10 years having a certificate indicating a medically identified ASD diagnosis and meeting the Childhood Autism Rating Scale (CARS) diagnostic criteria; (2) no physical or visual–auditory disability impairment; and (3) no medication or other motor intervention. As a result, 24 children with autism (7 females and 17 males) were included in the current report. Based on the CARS diagnostic criteria detailed in Section 2.2.2, and considering gender distribution, the 24 participants were divided into an experimental group (9 males and 3 females) and a control group (8 males and 4 females) using a matching method. One child in the experimental group dropped out of school due to long absences (>5 times), and two children in the control group dropped out due to incomplete testing. Thus, data from 21 children with ASD (9 males and 2 females in the experimental group; 7 males and 3 females in the control group) were included in the final analysis.

### 2.2. Experimental Implementation

#### 2.2.1. Procedure 

Following informed consent from parents/guardians, an orientation was conducted by the principal instructor to outline the plan of the program. Prior to commencement, an initial meeting involved all 24 children, their parents, and instructors, with a questionnaire being distributed to collect demographic and health history data, including clinical diagnoses and education history, alongside motivation strategies, during which, assent to the program was obtained from the parents of the children. Diagnostic and assessment tools like Autism Diagnostic Interview-Revised (ADI-R) were utilized for confirming diagnoses and assessing participants at the program’s start and end [13,14]. Parents/guardians received recommendations for ongoing physical activity and a behavioral assessment summary. Instructor training included ethics, CPRT, reinforcement strategies, and the importance of individualized instruction, with practical CPRT implementation examples. The adherence of the program to CPRT principles was monitored by the last two authors through session observations. Before each session, a 15 min debriefing and a 30 min discussion post-session were scheduled daily. The primary instructor reviewed the goal of the session and highlighted key words for target skills during debriefings, ensuring all instructors were prepared to demonstrate these skills. Post-session discussions involved the team reflecting on strategy implementations and planning improvements for future sessions [13,14].

The intervention program was implemented without interfering with normal teaching (e.g., language, moral rules, mathematics, and drawing and crafts), and the daily school teaching was identical between the two groups. The program contents for the experimental group and the control group are shown in Appendix A, respectively. Fundamental motor skill intervention content, including locomotor skills (e.g., running, hopping, and sliding) and object control skills (e.g., catching, kicking, and batting), is summarized in Appendix A.

The exercise time for both the experimental and control groups was 15:30–16:30 from Monday to Thursday (12 weeks, 60 min/session, 4 times/week). During the intervention, the experimental group received both the exercise prescription and the daily school teaching activities, while the control group only received the daily school teaching activities. The experimental group and the control group had the same number of instructors. In each intervention, there was a bishop and three teaching assistants. The bishop implemented the specific plan, and the teaching assistants assisted in completing the learning tasks, such as picking up children before and after class, arranging equipment, guiding students to fixed positions, and helping them learn the movement. In particular, one-on-one help and peer demonstration guidance were provided for children with poor understanding ability during the intervention process.

Exercise intensity was monitored by heart rate monitoring and the Physical Activity Scale for Children (PASC) after each exercise session during the intervention [31]. The PASC is designed specifically for children and adolescents aged 8 to 15 years, and the activities included in the scale are typical of those engaged in by children within this age range, ensuring that the questions are relevant and understandable for the target population. It is the PASC that has been extensively validated in the literature for its reliability and validity in measuring physical activity levels in children [31,32,33]. In addition, the children’s breathing, sweating, and mental status were observed during exercise participation, aiming to assist in controlling moderate-intensity exercise. The average exercise heart rate was maintained in the range of 50–75% of the maximum heart rate, so as to control exercise intensity at a moderate level. No child in the study participated in similar programs outside the classroom. The intervention implementation environment was determined to be indoors or outdoors according to weather conditions, and it was mainly outdoors.

The adopted CPRT method provides a natural teaching environment and motivation levels to promote the development of core skills in children with ASD, which might have a derivative effect on other behaviors [17,23]. In order for children to execute instructions correctly, the use of instructions complied with the following rules: (1) Verbal instructions should be accompanied by physical actions, and explanations should emphasize only the key points; (2) the instructions are mainly words, phrases, or whistles, and the language guidance should be expressed without sentences as much as possible; (3) the same action must use the same instruction, that is, the same instruction corresponds to the same action; (4) bright-colored equipment is used to interest the children; (5) the preparation part and the end part of each training class are completed with music; (6) the use of reinforcers should be selected according to the characteristics of the child, with positive reinforcement being the main one and negative reinforcement being the supplement, which does not mean that negative reinforcement cannot be used; (7) attention should be paid to the timing of using the reinforcers; (8) the process of reinforcement to de-reinforcement should be emphasized in the teaching process. In addition, the teaching process attaches importance to demonstration, body lift display, progressive time-delay prompts, and other strategies.

Moreover, based on the Newell triangle action constraint model [24], the activity-related factors were adjusted by increasing or reducing the difficulty of the content according to children’s ability level, aiming to ensure that all children could complete the same activity content (Table 1). Additionally, the strategies of how to apply and change the difficulty of content by taking the overhand toss as an example are illustrated in Appendix A, and for other interventions, they were carried out following the same strategies.

To ensure the quality of the intervention, the teachers cooperated with the adaptive sports team, and the scientific research team was responsible for guidance. The testers were trained before the tests, and the score of each skill test item required 90% agreement between the two assessors before it was recorded. 

#### 2.2.2. Measurement

##### Diagnostic Scale for ASD

The CARS compiled by Schoplen (1980) [25,31] consisted of 15 items with detailed assessment rules. The scale was scored on four levels, and each level was graded using the following criteria: (1) age-appropriate behavioral performance, (2) mild abnormality, (3) moderate abnormality, and (4) severe abnormality. The total score of the scale was 60 points; <30 points was considered non-autistic; 30–36 points and fewer than five items below 3 points was considered mild to moderate autism; ≥36 points and at least five items above 3 points was considered severe autism. 

##### Physical Activity

The ActiGraph GT3X+, a triaxial accelerometer, has been the most frequently used objective measurement instrument in recent studies on the physical activity levels of children with autism [32,33]. Its reliability and validity have been widely validated both domestically and internationally, including in the monitoring of physical activity in children with autism, intellectual disabilities, and other special needs [32,33,34,35]. This study utilized the ActiGraph GT3X+ accelerometer as the measurement tool for assessing the physical activity levels of children with autism. In the measurement, it was worn on the right upper iliac bone 3 times at the start, midpoint, and end of the study period, each lasting 7 consecutive days (including 5 working days), except for sleeping, bathing, swimming, and other water activities. Before wearing the accelerometer, the parents of the test subjects were trained on the requirements and precautions for wearing the accelerometer and were reminded about the accelerometer daily. Then, 1 day after the end of the test (i.e., the 8th day), Actilife 6.11.5 software was employed to export the data [34,35,36]. Based on Chinese children and adolescents, the threshold value of the accelerometer was selected as follows: sedentary activity, SA, 0~100 times/min; moderate-to-vigorous physical activity, MVPA ≥ 2800 times/min [35]. Based on the threshold value of the accelerometer, sedentary time was defined as periods of time during which participants were engaging in sedentary activity, such as sitting still, or engaging in low-intensity activities, which recorded using the ActiGraph GT3X+ accelerometer [35,36]. From physical activity behavior habits, children with autism and poor exercise ability have a higher frequency of sudden and unstable movements and spend a shorter time engaging in high-intensity physical activities [2]. Therefore, the sampling interval chosen in this study was 1 s to ensure that all physical activity behaviors were collected. After obtaining the results of the accelerometer data, the recommended reference amount was no less than 60 min of moderate-intensity physical activity per day [15]. 

##### Health-Related Fitness

According to the American Sports Medicine Association, health-related fitness is divided into four aspects: body composition, cardiorespiratory function, muscle strength and endurance, and flexibility [35]. Combining relevant research on evaluating children with ASD and the situation of neurotypical children, we selected the following indicators for four aspects: body mass index (BMI), the 20 m PACER test, muscle strength, and endurance. The reliability and validity of these indicators have been extensively studied for assessing body composition and predicting health outcomes [35]. 

Body Composition: Body composition was primarily assessed through BMI, calculated as weight (kg)/height (m)^2^. Height and weight were measured using a stadiometer (SH-8053, Jiangsu Suhong Medical Instruments Co., Ltd., Changzhou, China) and a scale (OMRON HBF-306, Omron Corporation, Kyoto, Japan), respectively, with participants in an upright position without shoes.

Waist-to-hip ratio: Waist measurement: In a standing position, with feet apart by 25–30 cm, we used a measuring tape to measure around the body, positioning the tape 2 cm above the navel. Hip measurement: While standing with legs straight and close together and arms naturally hanging by the sides, we placed the measuring tape horizontally at the level of the pubic bone and the most prominent part of the buttocks to measure around.

Grip strength: Grip strength, an indicator of upper limb static strength, reflecting forearm and hand muscle grip ability, was measured using an electronic handgrip dynamometer (model 4601a, Takei Kiki Kogyo Co., Niigata, Japan). Participants stood with their arms at a defined angle to the torso, with their palms facing inward and elbows fully extended, gripping the dynamometer with maximum effort using the dominant hand. Measurements were taken twice, and the highest value was recorded.

Sit-Ups: The number of sit-ups performed in one minute reflects abdominal muscle strength and endurance. Participants performed as many sit-ups as possible in one minute on a mat, with knees bent at 90 degrees and arms crossed over the chest. The test counted the number of complete sit-ups.

Vertical Jumps: We assessed the lower limb explosive power during vertical jumps using a vertical jump meter (model T.K.K 5406, Takei Scientific Instruments Co., Ltd., Niigata, Japan). Participants jumped as high as possible from the mat, measured over two attempts and recorded in centimeters.

PACER Test: This test is an indication of aerobic endurance. Participants ran between two points 20 m apart, pacing with audio cues. The test ended when the participant failed to reach the endpoint in time twice consecutively. Due to cognitive limitations, professional testers and physical education teachers assisted participants, ensuring clear instructions and encouragement.

Sit and Reach: This is a flexibility measurement performed using a sit and reach box (TAKEI, T.K.K. 5412, Tokyo, Japan). Participants sat against a wall with legs straight, heels together, and toes apart, reaching forward along the measuring board. The best of two attempts was recorded in centimeters.

Follow-up tests: We conducted follow-up tests 1 month after the intervention to assess the sustainability of the intervention effects. All participants underwent the same assessments during the follow-up as they did at the end of the intervention. This included re-measuring all the indicators mentioned above. Follow-up data were used to analyze the retention of the intervention effects and the long-term impact.

During the test, the study adjusted the test procedure and tasks: (1) In terms of equipment, brighter-colored materials were used to attract the test subjects’ attention, and directional arrows were painted on the test site to make the test subjects more aware of the kicking tasks; (2) in terms of movement explanations, the researcher increased the number of movement demonstrations and reduced the number of verbal explanations that contained overly long statements; (3) in terms of testers’ assistance, since children were more emotionally dependent on their teachers and were more familiar with the instructions given by their teachers and could carry their instructions out easily, the test subjects’ physical education teachers, in addition to the two testers, were present during the tests to assist with the test task. Before the formal start of the test, two or three practice sessions with physical and verbal assistance were conducted to help them understand the test task. Additionally, the testers verbally encouraged the children during the test and used appropriate reinforcers to ensure that the children could complete the test to the best of their ability.

### 2.3. Statistical Analysis

Statistical analysis was performed using SPSS 22.0. First, the Kolmogorov–Smirnov test was employed to check whether the obtained data were normally distributed, and then, the descriptive statistics were expressed as means ± standard deviation. Paired-sample *t*-tests were used to evaluate whether there were differences before and after the intervention, with the set statistical significance level of 0.05. The effects of the fundamental movement skill intervention on physical activity in children with ASD were analyzed using the repeated-measure analysis of variance (ANOVA).

## 3. Results 

### 3.1. Demographic Characteristics of the Participants

The demographic characteristics of the two groups of children with ASD were analyzed using independent-sample *t*-tests. The results showed that there was no significant difference (all *p* values > 0.05) between the two groups for age, height, weight, and BMI (Appendix A). According to the results of the C-PEP 3 test, there was 1 high-functioning child and 10 low-functioning children with autism in the experimental group and 2 high-functioning and 8 high-functioning children with autism in the control group.

### 3.2. Physical Activity Levels after Fundamental Movement Skill Intervention

The effects of the fundamental movement skill intervention on physical activity in children with ASD were analyzed using repeated-measures ANOVA. Physical activity at all levels was considered the dependent variable, where time (before and after the intervention) and group (children who received the intervention vs. children who did not) were considered independent variables. Figure 1 shows that time and group have interactive effects on pre- and post-test sedentary behavior (F(1,19) = 4.581, *p* = 0.01, partial η^2^ = 0.194), but not for the time engaged in moderate-to-vigorous physical activity (F(1,19) = 2.137, *p* = 0.719, partial η^2^ = 0.101).

Figure 2 shows the SB and MVPA obtained from the follow-up tests. It can be seen that there was a continually significant effect of the fundamental movement skill intervention on the time spent engaging in sedentary behavior (F(1,19) = 3.745, *p* = 0.02, partial η^2^ = 0.327), but not for the time engaged in moderate-to-vigorous physical activity (F(1,19) = 1.824, *p* = 0.847, partial η^2^ = 0.371). It was revealed that the experimental group showed continuous decreases in sedentary behavior from the pre–post-tests and from post- to follow up tests. In addition, the follow-up tests on sedentary behavior showed that values were maintained at the post-test levels for the experimental group.

### 3.3. Anthropometric Characteristics of Children with ASD before and after the Fundamental Movement Skill Intervention

The effects of the fundamental movement skill intervention on health-related fitness components among children with ASD were analyzed by repeated-measures ANOVA. The components of health-related fitness were considered dependent variables, where time (before and after intervention) and group (children who received the intervention and children who did not) were considered independent variables. There were statistically significant main effects of the groups before vs. after on the pre- and post-intervention body composition (Figure 3 and Appendix A), including BMI (F(1,19) = 8.631, *p* = 0.03, partial η^2^ = 0.312), but not waist-to-hip ratio (F(1,19) = 11.556, *p* = 0.184, partial η^2^ = 0.378). For cardiopulmonary function, there was no significant main effect of the groups before vs. after the intervention for the 20 m PACER fitness test (F(1,19) = 10.811, *p* = 0.531, partial η^2^ = 0.363). For muscle strength and endurance, significant main effects of the groups before vs. after the intervention were found for the vertical jumps (F(1,19) = 5.309, *p* = 0.04, partial η^2^ = 0.218) and sit-ups (F(1,19) = 3.376, *p* = 0.02, partial η^2^ = 0.151), but not for grip strength (F(1,19) = 14.495, *p* = 0.356, partial η^2^ = 0.433). In terms of flexibility, there was a significant main effect of the groups before vs. after the intervention for the sit and reach test (F(1,19) = 36.228, *p* = 0.02, partial η^2^ = 0.656).

Figure 4 shows the summarized the body composition, muscle strength and endurance, and flexibility results obtained from the follow-up tests. It is obvious that there was a continually significant effect for sit-ups (F(1,19) = 4.215, *p* = 0.01, partial η^2^ = 0.426), but not for body composition, including BMI (F(1,19) = 7.341, *p* = 0.621, partial η^2^ = 0.427) and waist-to-hip ratio (F(1,19) = 10.427, *p* = 0.157, partial η^2^ = 0.217); cardiopulmonary function, including the 20 m PACER fitness test (F(1,19) = 8.744, *p* = 0.278, partial η^2^ = 0.287); muscle strength and endurance, including vertical jumps (F(1,19) = 4.287, *p* = 0.897, partial η^2^ = 0.368) and grip strength (F(1,19) = 8.472, *p* = 0.628, partial η^2^ = 0.273); or flexibility, including the sit and reach test (F(1,19) = 18.745, *p* = 0.761, partial η^2^ = 0.463).

## 4. Discussion

This study evaluated the acquisition and retention effect of a 12-week fundamental movement intervention on the physical activity and health-related fitness of children with ASD. It demonstrates that the interventions produced significant changes in the sedentary time. Health-related physical fitness indicators showed that body composition significantly decreased body mass index and muscle strength (vertical jumps and sit-ups) and increased flexibility (sit and reach). The 1-month follow-up test results indicated that after the interventions, sedentary time continuously decreased, and muscle strength (sit-ups) significantly increased. There were no adverse reactions during the 12-week intervention. 

Consistent with the findings of established studies, sedentary time in children with ASD can be significantly improved after intervention with fundamental movement skills [36,37]. In addition, we found that the experimental group of children with ASD showed a sustained reduction in sedentary time, whereas there was no significant difference in the control group [15,36]. Interestingly, before the follow-up tests, neither the experimental nor the control group children participated in other school programs during the 1-month vacation. The sedentary time of children with ASD continuously decreased. The reason for this result may be ascribed to the promotion of motor participation in children with ASD through CPRT instruction during the intervention, which might also be related to the increase in positive emotions after the intervention [9,26,37]. Studies have shown that positive mood effects and self-efficacy are drivers of children’s motor participation, and that basic motor skills interventions can significantly increase positive mood and self-efficacy in children with ASD [38]. The sustained effects produced after the intervention not only highlight that CPRT instruction facilitates the acquisition of children’s motor skills, but also produces positive outcomes. Further, children’s participation in cooperating with multiple learning tasks and games during the implementation of the intervention led to more positive emotional experiences in children with ASD. Therefore, fundamental movement skill interventions are critical to the physical activity of children with autism. CPRT provides evidence-based, practice-based verbal and visual cues and feedback strategies to create the optimal learning environment for children with ASD, promote their motor participation, and ultimately reduce their static physical activity time.

However, these children did not show significant differences in MVPA, and we suggest that seasonal and climatic changes may be responsible for the reduced initiative and motivation of children to participate in physical activity [39]. In comparison with other seasons, in the winter, there is a trend of a significant decrease in the level of physical activity due to lower temperatures, including for unique populations [40]. As the temperature drops, the level of physical activity of both typical and unique groups decreases [41,42]. Akande et al. evaluated the physical activity of adults in summer and winter, and the results showed that the physical activity in winter was significantly lower than that in summer [43]. Thus, short-term interventions may not positively influence the physical activity patterns of children with autism, leading to no significant changes in the MVPA level of important factors.

During the growth and development of children, monitoring body composition indicators is essential. In this study, the indicators of body composition observed are BMI and the waist-to-hip ratio [44]. Our findings revealed a significant reduction in body mass in the experimental group post-intervention, despite the absence of dietary changes, aligning with previous research results. This reduction may be attributed to the children in the experimental group engaging in fundamental movement skills at a certain exercise intensity. This intervention can elevate lipase activity, promote fat hydrolysis, and facilitate the rapid oxidation of free fatty acids for energy. Consequently, this enhances fat mobilization and oxidative catabolism, reducing overall fat accumulation. Additionally, it increases the levels of UCP3 mRNA and catecholamines in skeletal muscle, thereby raising the basal metabolic rate and energy consumption [45,46,47]. However, there was no significant change in the waist-to-hip ratio in the experimental group, potentially due to the duration of the intervention. Overall, children with ASD showed significant improvements in body composition through targeted exercises. Nonetheless, these findings suggest that a longer intervention period may be necessary for more substantial effects.

In terms of cardiorespiratory fitness, studies have shown that good cardiac function is necessary for individuals to be able to easily complete their daily studies, live their lives, and perform exercises such as walking and running [48]. For children with typical development, only with good cardiac function can they participate in physical activities and increase the amount of time they spend exercising to gain more health benefits, feel relaxed and happy during long periods of play and interaction with peers, and improve their social skills. This is even more important for individuals with autism. Bricout et al. showed that cardiorespiratory fitness was lower in children with autism than in children with typical development [49]. The present study showed no significant change in the cardiorespiratory fitness index of the children in the experimental group after 12 weeks of intervention. The reason for this may be related to the fact that a variety of sports such as running, jumping, and ball games in the intervention program are still difficult for children with ASD. Meanwhile, cardiorespiratory fitness did not change significantly before and after the intervention, as influenced by the duration of the intervention. In addition, the school’s adherence to the safety-first philosophy had an impact on the sports participation of these children [50,51,52], a result that was also reflected in the follow-up tests.

At the same time, good muscle strength and muscular endurance have important implications for health promotion and injury prevention [42,53,54]. From the changes before and after the 12-week intervention, it was found that the performance of children with ASD improved in both the experimental and control groups, but the experimental group improvement was more pronounced in the experimental group. For children with ASD, the intervention process requires the movement of large and small muscles in the whole body, as well as an appropriate amount of physical activity and time, in order to achieve the effect of training muscle strength and muscular endurance, through the teaching of key responses to enable these children to participate in the process of uninterrupted cyclic movements, which require the coordination of all parts of the body, and as body movements which implicate the operation of different muscle groups [55], which is consistent with the research by Arslan [19] and Ketcheson [9,26].

In terms of flexibility, it is specific to joints and can be improved by regular stretching exercises [56]. The results of this study showed that after 12 weeks of intervention, the children with ASD in the experimental group showed a significant improvement in the sit and reach test, which was related to static stretching in the relaxation session of each intervention, and the regular stretching exercises throughout the intervention process had a positive effect on the improvement of joint flexibility [46,57].

## 5. Study Strengths, Limitations, and Future Directions

The present study examined the potential impact of fundamental movement skills on physical activity and health-related fitness in children with autism, providing actionable and regular fundamental movement skill programs for schools as an effective way to achieve the healthy development goals of children with autism.

However, due to the influence of subjective (e.g., participant adherence and self-report bias) and objective factors (e.g., sample size limitations, measurement tool constraints, and the short study timeframe), the following main limitations still exist: (1) Due to the large individualized differences in children with autism, the sample size selected for this study was small, which can explain the intervention effects to a certain extent, but the conclusions drawn may have limitations, and later studies should be based on the characteristics of Pei Chi schools and fully mobilize peer resources for extensive and in-depth research. (2) The selection of children with autism was concentrated in the city. Therefore, the children with autism in this study may have enjoyed a higher social status and more family support compared with other children with autism of the same age, and further exploration of the development of children with autism in different cultural contexts is still needed at a later stage.

## 6. Conclusions

This study contributes to a small but growing body of research exploring the effectiveness of physical activity interventions for children with ASD. It highlights the significance of examining the sustained effects of these interventions, emphasizing improvements related to the study and program planning. Our findings indicate an overall enhancement in fundamental movement skills following a CPRT-based physical activity program. Specifically, the sedentary time was significantly decreased, and there were large changes in health-related physical fitness indicators, including significantly improved body composition, muscle strength and endurance, and flexibility. Further, follow-up testing revealed differences in maintenance, including that the sedentary time of children continued to reduce, and the muscle strength and endurance improved continuously after the intervention. In summary, the fundamental movement skill intervention helped to decrease the sedentary time and contributed to health-related fitness in children with ASD. In the future, schools can provide actionable and regular fundamental movement skills programs for children with autism as an effective way to achieve children’s healthy development goals.

## Figures and Tables

**Figure 1 healthcare-12-01304-f001:**
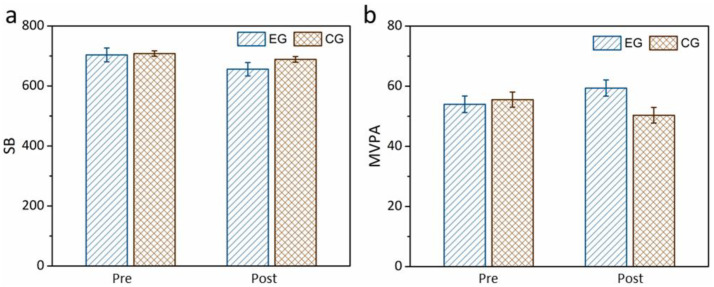
Pre–post comparisons on the (**a**) SB and (**b**) MVPA. Abbreviations: SB, sedentary behavior; MVPA, moderate-to-vigorous physical activity; EG, experimental group; CG, control group.

**Figure 2 healthcare-12-01304-f002:**
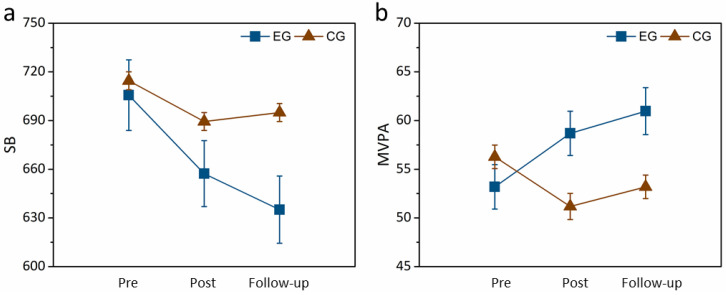
Follow-up test on the (**a**) SB and (**b**) MVPA. Abbreviations: SB, sedentary behavior; MVPA, moderate-to-vigorous physical activity; EG, experimental group; CG, control group.

**Figure 3 healthcare-12-01304-f003:**
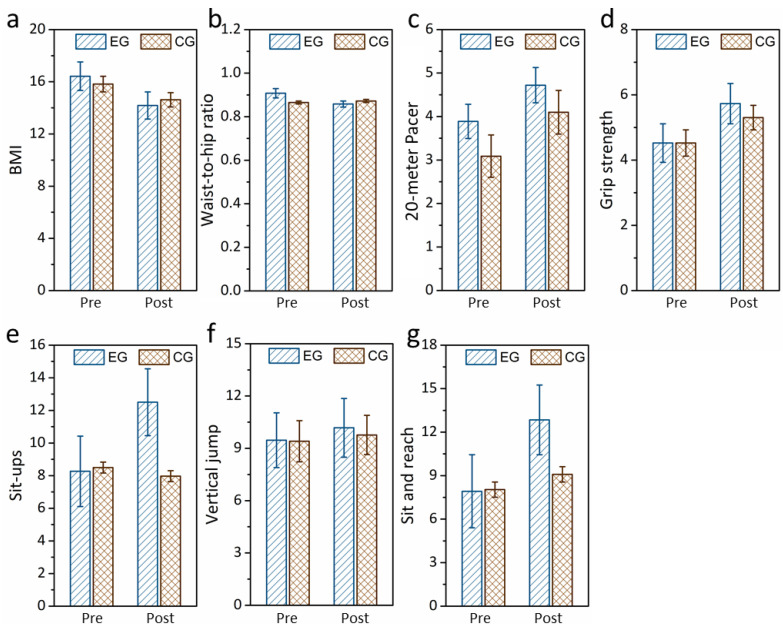
Pre–post comparisons of body composition, including (**a**) BMI and (**b**) waist-to-hip ratio; (**c**) cardiopulmonary function, including 20 m PACER; muscle strength and endurance, including (**d**) grip strength, (**e**) sit-ups, and (**f**) vertical jumps; flexibility, including (**g**) sit and reach test.

**Figure 4 healthcare-12-01304-f004:**
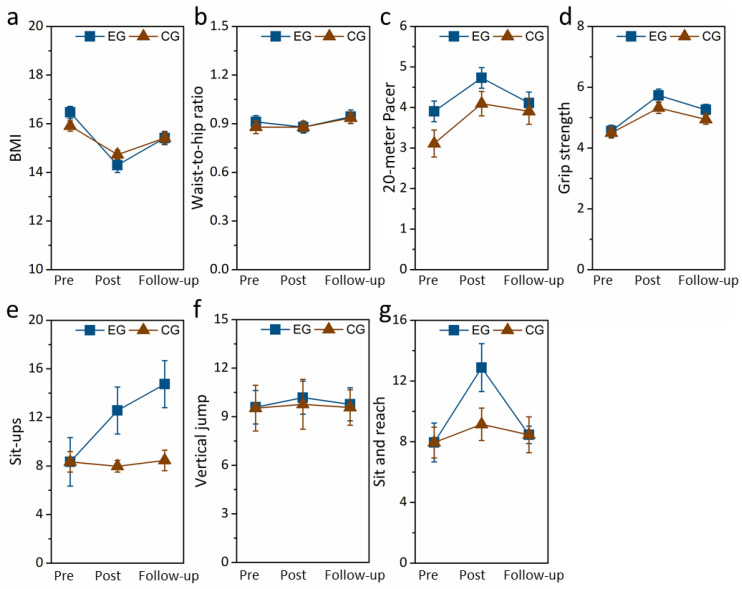
Follow-up tests on body composition, including (**a**) BMI and (**b**) waist-to-hip ratio; cardiopulmonary function, including (**c**) 20 m PACER test; muscle strength and endurance, including (**d**) grip strength, (**e**) sit-ups, and (**f**) vertical jumps; flexibility, including (**g**) sit and reach test.

**Table 1 healthcare-12-01304-t001:** Applied difficulty of contents affected by changes in fundamental movement skill constraints.

Activity Constraint	Easy	Moderate	Difficulty
Orientation	Autochthonous	Move back and forth	Move side to side
Route	Straight line	Z-line	Circle
Speed	Slow	Fast	Combination of fast and slow
Distance	Short distance	Medium distance	Long distance
Strength	Bare-handed	Wear sandbags around one’s legs	Leg and hand sandbags
Restriction condition	Upper or lower limbs	Upper limb fit	Lower limbs unchanged; upper limbs changed
Equipment	Walk a 20 cm wide balance beam	Walk a 15 cm wide balance beam	Walk a 10 cm wide balance beam
Site	Level ground	Sloping ground	Spanned ground

## Data Availability

The data presented in this study are available on request from the corresponding author.

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
