# Peer review of "Acquisition and Retention Effects of Fundamental Movement Skills on Physical Activity and Health-Related Fitness of Children with Autism Spectrum Disorder"

_healthcare, 2024, doi:10.3390/healthcare12131304_

Round 1

Reviewer 1 Report

Comments and Suggestions for Authors

Thank you for the opportunity to review your study. The current investigation addressed an important topic regarding a fundamental movement skills training intervention on physical activity and health-related fitness of children with ASD in a Chinese sample. I have a few suggestions that I hope you find useful:

Abstract and Introduction

Line 18: Please report the actual p-value.

Line 25: The phrase "school in the future" sounds too ambiguous, as the evidence is based on a small sample from China. Consider rephrasing to be more specific.

Line 32: Please add "the" before "US" and change "A" to lowercase.

Line 99: Please ensure consistent use of the abbreviation "ASD" throughout the manuscript.

Line 118: Given the nature of this study, please address the research questions or hypotheses at the end of the introduction section.

Materials and Methods

Line 132: Did the authors conduct a power analysis to determine the sample size? Please provide details.

Line 135: Please clarify the group assignment method used in the study (e.g., randomization, matching).

Line 175: Please consider providing age-appropriateness and validation evidence regarding the Physical Activity Scale for Children.

Line 293. Please clarify the reason for applying for the t-test as there are two groups being tested triple times. Additionally, did the authors control for any confounding variables? If so, ANCOVA might be an appropriate choice.  

The authors have done a good job presenting the test results. However, given the significant interactions, were any post-hoc analyses conducted? If so, please report the results; if not, consider performing post-hoc tests to further explore the differences between groups and time points.

In addition, from an ethical perspective, did the control group have a chance to receive the intervention afterwards? Providing beneficial intervention to the control group after the study would be a considerate and ethical approach. 

Discussion

Line 404. This is an interesting point, and it is supported by many studies using an ecological framework, which suggests that macro-system factors like temperature indeed influence physical activity in children.

Reviewer 2 Report

Comments and Suggestions for Authors

Comments on the Quality of English Language

Minor edits needed. 

Round 2

Reviewer 2 Report

Comments and Suggestions for Authors

Thank you for all your hard work in this manuscript. The manuscript is now much improved. Congratulations.